# The Challenges of Patient Selection for Prostate Cancer Focal Therapy: A Retrospective Observational Multicentre Study

Alessio Paladini [1], Giovanni Cochetti [1,*], Alexandre Colau [2], Martin Mouton [2], Sara Ciarletti [1], Graziano Felici [1], Giuseppe Maiolino [1], Federica Balzarini [2], Philippe Sèbe [2] and Ettore Mearini [1]

1 Urology Clinic, Department of Medicine and Surgery, University of Perugia, 06129 Perugia, Italy
2 Chirurgie Urologique, Hôpital Croix Saint Simon Groupe Diaconesses, 75020 Paris, France
* Correspondence: giovanni.cochetti@unipg.it

**Abstract:** Increased diagnoses of silent prostate cancer (PCa) have led to overtreatment and consequent functional side effects. Focal therapy (FT) applies energy to a prostatic index lesion treating only the clinically significant PCa focus. We analysed the potential predictive factors of FT failure. We collected data from patients who underwent robot-assisted radical prostatectomy (RARP) in two high-volume hospitals from January 2017 to January 2020. The inclusion criteria were: one MRI-detected lesion with a Gleason Score (GS) of $\leq 7$, $\leq$cT2a, PSA of $\leq 10$ ng/mL, and GS 6 on a random biopsy with $\leq 2$ positive foci out of 12. Potential oncological safety of FT was defined as the respect of clinicopathological inclusion criteria on histology specimens, no extracapsular extension, and no biochemical, local, or metastatic recurrence within 12 months. To predict FT failure, we performed uni- and multivariate logistic regression. Sixty-seven patients were enrolled. The MRI index lesion median size was 11 mm; target lesions were ISUP grade 1 in 27 patients and ISUP grade 2 in 40. Potential FT failure occurred in 32 patients, and only the PSA value resulted as a predictive parameter ($p < 0.05$). The main issue for FT is patient selection, mainly because of multifocal csPCa foci. Nevertheless, FT could represent a therapeutic alternative for highly selected low-risk PCa patients.

**Keywords:** prostate cancer; focal therapy; low-risk PCa; MRI; PSA

## 1. Introduction

Prostate cancer (PCa) is the second most common male cancer with the highest incidence in Western countries, which is partially due to the wide use of prostate-specific antigen (PSA) as the primary screening tool [1,2]. On the one hand, if the screening based on PSA has been associated with a decrease in PCa-related mortality, on the other hand, it leads to over-diagnosis and over-treatment of silent PCa at the expense of functional side effects [3]. In the era of precision medicine, the goal is to diagnose and treat only clinically significant PCa (csPCa), thus individualizing the treatments on the patient's disease characteristics.

Recent studies focused on the value of miRNAs as new tools for early cancer diagnosis, but this goal has yet to be reached [4]. Multiparametric magnetic resonance imaging (MRI) has been improving the diagnostic algorithm of csPCa; it is recommended before any prostatic biopsy, and it has been included in the protocols of active surveillance (AS) and active monitoring (AM) for PCa [2,5]. An MRI-target biopsy could improve the detection rate of ISUP grade $\geq 2$ and ISUP grade $\geq 3$, approximately 40% and 50%, respectively. For this reason, the EAU guidelines recommend performing naïve biopsy combining the target procedure with a systematic one [2].

Once PCa has been diagnosed, an individualised treatment respecting oncological outcomes is mandatory. For low-risk localised PCa, AS/AM is advisable for a well-informed patient who accepts the risk of progression and a strict follow-up protocol. For all the others, except the locally advanced, a local treatment, if feasible, should be proposed in

personalised medicine [2,6]. Unfortunately, radical prostatectomy, even if nerve-sparing and radiotherapy, are not free of side effects [7,8]. For surgery, incontinence and erectile dysfunction rate are reported to be 31% and 38% at 12 months, respectively, if nerve-sparing techniques are performed [9,10]. Neither open, laparoscopic nor robotic approaches have demonstrated a clear superiority compared to others in terms of functional outcomes and quality of life [11,12]. Even in the case of minimally invasive surgery, patients could suffer from intra-, peri- and post-operative complications [13]. For radiotherapy, the sexual and incontinence domains show a similar result to surgery, but gastrointestinal adverse effects are predominant [2,14]. An altered functional state could impair the health-related quality of life (HRQOL), irrespective of treatment choice [15].

In particular, functional results are of paramount importance in patients who underwent radical prostatectomy for localised low- and intermediate-risk PCa because these outcomes are more emphasised. For these risk categories, focal therapy (FT) has gained increasing importance worldwide in recent years, offering an alternative to whole-gland treatment. Focal therapy (FT) for PCa applies energy to an index lesion and its surrounding margins to treat only clinically significant lesions preserving neurovascular structures close to the prostate gland [16]. The advantages of FT are exciting functional outcomes in terms of urinary continence, sexual potency and quality of life with reasonable short-term oncological safety [16]. However, FT is recommended only within the setting of clinical trials using predefined inclusion criteria and scheduled surveillance [17,18]. Population-based studies evaluating FT as a treatment for PCa are poor. Recently, Flegar et al. analysed FT cases in Germany and reported an increase from 2006 to 2008 and then a decrease until 2014. Since 2015, the overall cases of FT have shown a plateau trend [16]. The early increase in the utilization of FT is due to the development of new technologies. In contrast, the reason for the subsequent decrease in case numbers is the significant risk for tumour recurrence or progression after FT [16,19].

The current PCa diagnostic algorithm could fail to propose a focal therapy because of the risk of missing other csPCa not revealed by MRI. In this retrospective observational population-based multicentre study on patients who underwent bilateral nerve-sparing RARP for localised low-risk PCa, we analysed the preoperative patients and the disease's characteristics as potential predictive factors of FT failure.

## 2. Materials and Methods

In this multicentre study, we prospectively collected data from patients who consecutively underwent robot-assisted radical prostatectomy (RARP) in two high-volume tertiary hospitals (Clinica Urologica—Università degli Studi di Perugia Azienda Ospedaliera di Perugia-Italy, Service de Chirurgie Urologique Groupe Hospitalier Diaconesses Hôpital Croix Saint-Simon de Paris-France) from January 2017 to January 2020.

We performed dosages of PSA, digital rectal explorations (DREs), multiparametric-3-Tesla-MRI, MRI/ultrasound fusion guided and systematic prostatic biopsies.

We collected clinicopathological data regarding age, pre-operative PSA, prostate volume, PSA-density, clinical T stage, MRI-index lesion diameter, MRI-index lesion Prostate Imaging—Reporting and Data System version 2 (PI-RADS v2) score, Gleason score (GS) of MRI-index lesion on biopsy, Gleason score by systematic biopsy, clinical International Society of Urologic Pathologists (ISUP) grade, Gleason score and ISUP grade on histology, extracapsular-extension, PSA at 40 days and 3-6-12 months after RARP, biochemical, local or metastatic recurrence within 12 months and time to recurrence.

Inclusion criteria included patients who underwent bilateral nerve-sparing RARP with negative surgical margins, with only one MRI-detected lesion, $\leq$cT2a, PSA $\leq$ 10 ng/mL, GS $\leq$ 7 on target biopsy, GS 6 on random biopsy with $\leq$2 positives foci out 12.

We assumed the safety criteria in terms of the inclusion criteria on the final specimen, no extracapsular extension, and no biochemical, local, or metastatic recurrence within 12 months. Based on this definition, patients were divided into two groups: the first one was the FT success group, which included patients matching all of the safety criteria;

the second was the FT failure group, which consisted of those who did not match the safety criteria.

## 3. Results

A total of 67 patients who underwent RARP with bilateral nerve-sparing technique, matching the inclusion criteria, were enrolled. Of these, 35 patients (52.2%) were included in the FT success group and 32 (47.8%) in the FT failure group according to the potential oncological safety criteria.

Thirty-five (52.2%) had a clinical stage T1c and 32 (47.8%) cT2a. The mean age at diagnosis was 64.4 ± 6.65 years, the mean BMI 25 ± 1.56 kg/m$^2$, and the mean prostate volume 46.1 ± 15.5 cc without significant difference between the FT success and FT failure groups. MRI index lesion diameter, PI-RADS value, and bioptic ISUP group did not differ significantly between groups. PSA density score differed between the groups (14.8 vs. 17.8 ng/mL$^2$) but did not reach the statistical significance ($p = 0.10$). The mean PSA value in the FT success group was 6.14 ± 2.26 ng/mL vs 7.44 ± 1.92 ng/mL in the FT failure group, $p = 0.01$. The mean sise of the index lesion on the multiparametric MRI was 11.6 ± 4.56 mm; in twelve (17.9%) patients, the index lesion was classified as PI-RADS 3, in 33 (49.3%) as PI-RADS 4, in 22 (32.8%) as PI-RADS 5. The ISUP grade of the MRI-target lesions was 1 in 27 patients (40.3%) and 2 in 40 patients (59.7%). Clinicopathological features of the study population are shown in Table 1.

**Table 1.** Clinicopathologic features of our sample.

| Variables * | Total ($n = 67$, 100%) | FT Success ($n = 35$, 52.2%) | FT Failure ($n = 32$, 47.8%) | $p$ |
|---|---|---|---|---|
| Age (years) | 64.4 (±6.65) | 64.4 (±6.64) | 64.3 (±6.77) | 0.95 |
| BMI (kg/m$^2$) | 25.0 (±1.56) | 25.1 (±1.72) | 24.9 (±1.39) | 0.61 |
| CCI score | 4 (5–3) | 4 (5–3) | 4.5 (5–3.25) | 0.36 |
| IPSS score | 10 (13–7) | 10 (14–7) | 9.5 (12–7.25) | 0.97 |
| Prostate volume (cc) | 46.1 (±15.5) | 47.1 (±16.7) | 45.0 (±14.2) | 0.57 |
| PSA (ng/mL) | 6.76 (±2.19) | 6.14 (±2.26) | 7.44 (±1.92) | *0.01* |
| PSA density (ng/mL$^2$) | 16.2 (±7.21) | 14.8 (±8.18) | 17.8 (±5.71) | 0.10 |
| MRI index lesion diameter (mm) | 11.6 (±4.56) | 11.8 (±4.41) | 11.3 (±4.82) | 0.65 |
| PI-RADS<br>PI-RADS 3, *n* (%)<br>PI-RADS 4, *n* (%)<br>PI-RADS 5, *n* (%) | 12 (17.9%)<br>33 (49.3%)<br>22 (32.8%) | 9 (75.0%)<br>16 (48.5%)<br>10 (45.5%) | 3 (25.0%)<br>17 (51.5%)<br>12 (54.5%) | 0.21 |
| Biopsy ISUP group<br>ISUP 1, *n* (%)<br>ISUP 2, *n* (%) | 27 (40.3%)<br>40 (59.7%) | 17 (63.0%)<br>18 (45%) | 10 (14.9%)<br>22 (55%) | 0.14 |

\* Continuous variables with normal distribution were presented as mean (±standard deviation (SD)); non-parametric categorical variables were reported as absolute and relative frequencies (*n*, %), and non-parametric numerical variables with median (interquartile range—IQR). In italic the significant *p*-value.

In the FT failure group, a second csPCa focus on the specimen was discovered in 31 patients (96.9%), an ISUP grade 3 in 8 patients (25%), ISUP grade 4 in 2 (6.25%), pT3 in 12 patients (37.5%), and biochemical recurrence in 2 patients (6.25%). The uni- and multivariate logistic regression analysis on failure therapy was performed. The PSA values of <6 and >7 ng/mL were shown to be the only predictor value of FT success and failure, respectively, both in the univariate and multivariate logistic regression analysis, Table 2. PSA value > 7 ng/mL showed a sensitivity of 62.5%, a specificity of 61.1%, and an AUC of 0.73.

**Table 2.** Uni- and multivariate logistic regression analysis on failure therapy.

| | Univariate | | | Multivariate | | |
|---|---|---|---|---|---|---|
| | **HR** | **CI (95%)** | *p* | **HR** | **CI (95%)** | *p* |
| Age (years) | 0.99 | 0.92–1.07 | 0.95 | 1.01 | 0.93–1.11 | 0.77 |
| BMI (kg/m$^2$) | 0.92 | 0.67–1.26 | 0.61 | 0.96 | 0.65–1.43 | 0.85 |
| CCI score | 1.25 | 0.79–1.95 | 0.33 | 1.48 | 0.87–2.54 | 0.15 |
| IPSS score | 1 | 0.95–1.11 | 0.96 | 1.01 | 0.90–1.15 | 0.80 |
| Prostate volume (cc) | 0.99 | 0.96–1.02 | 0.57 | 0.95 | 0.86–1.04 | 0.27 |
| PSA (ng/mL) | 1.35 | 1.05–1.74 | *0.02* | 1.97 | 1.00–3.89 | *0.04* |
| PSA density (%) | 1.06 | 0.98–1.14 | 0.10 | 0.86 | 0.66–1.12 | 0.27 |
| MRI index lesion diameter (mm) | 0.98 | 0.88–1.09 | 0.65 | 0.94 | 0.80–1.10 | 0.45 |
| PI-RADS PI-RADS 4 vs. PI-RADS 3 | 3.19 | 0.73–13.9 | 0.12 | 2.47 | 0.49–12.3 | 0.27 |
| PI-RADS 5 vs. PI-RADS 3 | 3.60 | 0.76–17.0 | 0.10 | 3.59 | 0.44–28.7 | 0.23 |
| Biopsy ISUP group (ISUP 2 vs. ISUP 1) | 2.08 | 0.77–5.64 | 0.15 | 1.23 | 0.33–4.49 | 0.76 |

In italic the significant *p*-value.

## 4. Discussion

The definition of FT is "ablation of a cancerous lesion, diagnosed by imaging and confirmed by biopsy, with a margin of safety surrounding the target lesion" [20]. Doubts persist about the correct definition of target lesion; among experts, there is a consensus that the focus is a lesion ISUP grade ≥ 2, but there is no consensus that the larger lesion is the target of FT [20]. If we could treat the index lesion with a safety margin, the functional results would be preserved, and the oncological ones respected [21]. Unfortunately, although FT for PCa has been studied for more than 10 years, little is known about this procedure's optimal technology and oncological safety [15,20].

Identification of the tumour lesion and the certainty of its focal nature, the destruction of the tumour lesion with an acceptable safety margin, and a careful follow-up to diagnose the persistence or disease relapse are cornerstones of FT success.

To reach an international consensus, Van den Bos et al. proposed, as eligible criteria for FT, to include PSA levels < 15 ng/mL, clinical stage T1c-T2a, ISUP 1 or 2, life expectancy > 10 years, and any volume of the prostate gland [22]. In our study, the inclusion criteria were more restricted, enrolling only patients with low-risk PCa according to EAU criteria and those with ISUP grade 2 in a maximum of two bioptic foci. This was done to identify a category of patients with disease characteristics immediately close to those treated by active surveillance and, therefore, would have benefited more from a focal treatment. Patients with a higher risk of progression and recurrence should not be recruited since they need a lymphadenectomy for proper staging and treatment.

In our recruitment, we excluded patients with more than two positive bioptic cores because of the higher risk of BCR at 3 and 5 years, as per Truesdale et al. [23].

Perlis et al. proposed FT in selected patients with GG3-5 disease, especially in those who had particular attention to the functional outcomes or in those who had contraindications for radical treatment, be it radical prostatectomy or radiation therapy [17]. Encouraging results were highlighted in a study with selected patients affected by non-metastatic csPCa treated by HIFU. The five-year metastasis-free survival in the intermediate and high-risk groups was 99% and 97%, respectively [24]. Johnson et al. are of the opposite opinion in treating ISUP grade > 3 patients, even if highly selected. According to the authors, these patients are at high risk of persistent disease after FT because MRI has low sensitivity for identifying individual tumours foci. Approximately 22% of all PCa GG 3–5

are not correctly identified and located in the corresponding quadrant by MRI, and this percentage increases to 30% among patients with multifocal PCa. Furthermore, MRI does not reliably rule out the presence of extracapsular extension, and its diagnostic capacity appears to decrease for higher-grade lesions. In support of this, Johnson et al. reported that in their clinical experience, 48% of patients who were potential candidates for prostatic hemiablation had bilateral clinically significant disease [25]. In fact, mpMRI could omit about 20–45% of clinically significant PCa [17,26,27]. In our study, 46.3% of the patients analysed had either a second csPCa focus not detected by mpMRI or the disease had spread to the entire lobe or both. This percentage is similar to those reported in the literature showing that the inter variability is very low in the hub centres.

In the literature, however, some cases have been successfully treated. That is the case of Linder et al., who reported their promising experiences on FT laser treatment of four patients, two of which with ISUP 3. The authors described the creation of a convergent ablation with the absence of viable cells in the treated regions verified by MRI 7 days post-treatment and confirmed in the histopathology piece, but long-term data were not reported [28].

An aspect that should not be underestimated is the discrepancy between the clinical Gleason score and the pathological one. In fact, on the definitive diagnosis, about 50% of cases of upgrading in low-risk patients and up to 80% of downgrading in patients belonging to the intermediate and high-risk categories [29]. In the literature, five studies have reported a positive predictive value of <60% for detecting low-risk PCa. Although this finding can be partly explained by the fact that all studies had the lowest prevalence of low grades, it remains a surprising value [29–33]. Schiffmann et al. reported an upgrading rate of 55% for patients eligible for active surveillance and 78% for low-risk patients not eligible for active surveillance. In the same categories of patients, they reported an upstaging rate of 8% and 15%, respectively [34]. These data confirm the study by Busch et al. in which the upgrading and upstaging were 53.1% and 12.2%, respectively, for the group in active surveillance [35]. In our study, the upgrading rate was 38.1%, and for low-risk PCa patients, it was 40.7%. The possible explanation is that hub centres for PCa have the lowest rate of low-grade prostate cancer.

The lack of certainty of the disease grade and stage is a limiting factor for FT due to the risk of not respecting the oncological goal. It is necessary to assess the correct PCa grade and stage to choose the best therapeutic option. In our case, the potential success was around 52.2% and was largely influenced by radical post-prostatectomy upstaging. That underlines that a disease close to the capsule could also be poorly controlled by FT due to the lack of a safety margin, if not at the expense of the vascular-nerve bundles and a greater risk on nearby structures.

In our study, the only variable that significantly differs between groups is the PSA value, which was higher in the FT failure group. The mean value of the failure group was 7.44 ng/mL, which is only a slightly higher value. Boniol et al. reported an increasing linear risk of PCa for each percentage increase in PSA level with an odd ratio of 1.079 (95% confidence interval); however, this association is not related to a more aggressive disease [36]. Park YH et al., in their study, showed that the risk of prostate cancer was higher in men with a fluctuating PSA level and PSAV $\geq$ 1.0 ng/mL/yr than in those with a fluctuating PSA level and PSAV < 1.0 ng/mL/yr [37]. Moreover, patients with a positive digital rectal examination (DRE) and higher PSA levels show a higher risk of contralateral disease and may not be ideal candidates for FT [38]. These findings could explain why a small PSA level change that we found in our study could be significant. PSA density, as well, was different between groups but without significance, most likely due to the small samples. The small sample size represents the main limitation of our study.

## 5. Conclusions

The main issue for FT is patient selection, mainly because of multifocal csPCa foci not detected with the current diagnostic tools.

Our findings show that serum PSA could predict the success or failure of FT using a cut-off of 6 and 7 ng/mL, respectively. Further studies with a larger sample size are needed to confirm our results.

**Author Contributions:** Conceptualization, E.M., A.P. and P.S.; Methodology, G.C. and A.P.; Software, G.M.; Formal Analysis, G.M. and G.F.; Investigation, S.C., A.C., M.M. and F.B.; Data Curation, A.C., M.M., F.B. and G.F.; Writing—Original Draft Preparation, A.P.; Writing—Review and Editing, G.C. and E.M.; Visualization, S.C.; Supervision, E.M. and P.S. All authors have read and agreed to the published version of the manuscript.

**Funding:** This research received no external funding.

**Institutional Review Board Statement:** The study was conducted in accordance with the Declaration of Helsinki. Ethical review and approval were waived for this study due to the use of the gold standard treatment for the disease according to the European Association of Urology Guidelines.

**Informed Consent Statement:** Written informed consent has been obtained from the patient(s) to publish this paper.

**Data Availability Statement:** The data presented in this study are available on request from the corresponding author. The data are not publicly available due to privacy.

**Conflicts of Interest:** The authors declare no conflict of interest.

## Abbreviations

PCa: prostate cancer, FT: focal therapy, GS: Gleason Score, PSA: prostate-specific antigen, csPCa clinically significant prostate cancer, MRI: magnetic resonance imaging, AS: active surveillance, AM: active monitoring, HRQOL: health-related quality of life, RARP: robot-assisted radical prostatectomy, PI-RADS v2: Prostate Imaging—Reporting and Data System version 2, ISUP: International Society of Urologic Pathologists, DRE: digital rectal examination.

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
