# Peer review of "The Challenges of Patient Selection for Prostate Cancer Focal Therapy: A Retrospective Observational Multicentre Study"

_curroncol, doi:10.3390/curroncol29100538_

Round 1

Reviewer 1 Report

In their multicenter retrospective study "The challenging patients’ selection for prostate cancer focal therapy: a retrospective observational multicenter study" Paladini et al. investigated possible patients for focal therapy of prostate cancer by studying 67 patients who underwent robotic assisted radical prostatectomy between 2017 and 2020. The key findings are interesting and relevant since FT is an emerging therapy option.

Minor points:

- “Abbreviations” section should be included

- Spell check for English language  e.g. (we collected data of patients underwent RARP , etc.)

- in the introduction part, more focus should be put on FT (add 2-3 sentences about trends/epidemiology etc.) -   I would suggest to include the following citation:

Flegar L, Zacharis A, Aksoy C, et al. Alternative- and focal therapy trends for prostate cancer: a total population analysis of in-patient treatments in Germany from 2006 to 2019. World J Urol. 2022;40(7):1645-1652. doi:10.1007/s00345-022-04024-0

- Results/ Discussion section are clear and understandable

Author Response

We thank the reviewers for their suggestions and comments which allowed us to improve the quality of the manuscript.

Reviewer 1:

Thank you for your valuable comments and suggestions.

  • Comment: “Abbreviations” section should be included

Reply: “Abbreviations” section was added

  • Comment: Spell check for English language e.g. (we collected data of patients underwent RARP, etc.)

            Reply: A deep spell check for English language was performed by a native speaker

  • Comment: in the introduction part, more focus should be put on FT (add 2-3 sentences about trends/epidemiology etc.) -   I would suggest to include the following citation: Flegar L, Zacharis A, Aksoy C, et al. Alternative- and focal therapy trends for prostate cancer: a total population analysis of in-patient treatments in Germany from 2006 to 2019. World J Urol. 2022;40(7):1645-1652. doi:10.1007/s00345-022-04024-0

Reply: We revised the “Introduction” section focusing on the trend of focal therapies,  their advantages and their limits. We included the suggested citation too.

Thank you for your consideration.

Sincerely,

Giovanni Cochetti, MD, Urologist

Reviewer 2 Report

In this retrospective study, the authors used the regression models to predict focal therapy (FT) failure of prostate cancer (PCa), which are even missed by MRI scans. FT applies energy to prostatic index lesions hence treating only the clinically significant PCa selectively. Note that PSA as a proxy for Pca can lead to overtreatment for prostate cancers that are not clinically significant. The authors looked at different clinicopathological factors affecting FT failure and found PSA levels to be the predictive marker for FT failure (or success). The PSA levels for FT success and failure are 6.14 (± 2.26) and 7.44 (± 1.92), respectively. The difference is not very drastic, though it is significant both in the univariate and multivariate analysis. The number of patients for this study is not that high, and further validation retrospectively and/or prospectively in larger cohorts of patients is needed. That being said, this study is interesting, uses the right statistics, and opens new doors for more studies and better diagnoses of prostate cancer.
-Scientific English and sentence structures could be improved overall. e.g., this line does not read well- Line 85: '...Safety criteria were defined as the respect of clinicopathological inclusion criteria...'
-Please include the number of patients in the abstract.-Authors need to cite other examples where even a small change in PSA levels, like in the current study, can lead to differential varied prognosis. 

Author Response

We thank the reviewers for their suggestions and comments which allowed us to improve the quality of the manuscript.

Reviewer 2

Thank you for your valuable comments and suggestions.

  • Comment: Scientific English and sentence structures could be improved overall. e.g., this line does not read well- Line 85: '...Safety criteria were defined as the respect of clinicopathological inclusion criteria...'

            Reply: As suggested, we improved the Scientific English language. A deep revision of the language has been performed by a native speaker

  • Comment: Please include the number of patients in the abstract.

Reply: The number of patients was included in the abstract

  • Comment: Authors need to cite other examples where even a small change in PSA levels, like in the current study, can lead to differential varied prognosis.

Reply: In the discussion section, we reported other studies showing small change in PSA levels can affect the prognosis.  “…Boniol et al. report that there is a linear increasing risk of PCa for each percent increase in PSA level with an odd ratio of 1.079 (95% confidence interval), however, this association is not related to a more aggressive disease. Park YH et al. in their study show that the risk of prostate cancer was higher in men with a fluctuating PSA level and PSAV ≥1.0 ng/mL/yr than in those with a fluctuating PSA level and PSAV <1.0 ng/mL/yr. Moreover, patients with a positive digital rectal examination (DRE) and higher PSA levels show higher risk of contralateral disease and may not be ideal candidates for FT. These findings could explain why a small PSA level change that we found in our study could be significant.”

Thank you for your consideration.

Sincerely,

Giovanni Cochetti, MD, Urologist
